# Combining Isotope and Hydrogeochemistry Methods to Study the Seawater Intrusion: A Case Study in Longkou City, Shandong Province, China

**Yuxue Wang** [1,2,3], **Juxiu Tong** [1,2,*] , **Bill X. Hu** [4] **and Heng Dai** [5]

1. School of Water Resources & Environment, China University of Geosciences (Beijing), Beijing 100083, China; x18811651516@163.com
2. MOE Key Laboratory of Groundwater Circulation and Environmental Evolution, China University of Geosciences (Beijing), Beijing 100083, China
3. Beijing Hydrology Center, Haidian District, Beijing 100089, China
4. School of Water Conservancy and Environment, University of Jinan, Jinan 250022, China; bill.x.hu@gmail.com
5. State Key Laboratory of Biogeology and Environmental Geology, China University of Geosciences, Wuhan 430078, China; daiheng@cug.edu.cn
* Correspondence: juxiu.tong@cugb.edu.cn; Tel.: +86-18210615615

**Abstract:** In order to study hydrogeochemical effect in the process of seawater intrusion (SI), and provide scientific basis for comprehensive management of water resources and water ecological restoration, the Longkou city of Shandong province in China was taken as an example in this study. Based on the observed data, traditional hydrogeochemistry methods of hydrochemistry analysis, correlation analysis, principal component analysis, and reverse geochemical simulation was firstly comprehensively combined with stable isotope tracing in Longkou city, and this is the first study to use the isotope method to study SI in the study area. The results showed $Cl^-$ had high correlation with $Na^+$, $Mg^{2+}$, and $K^+$. The hydrochemical types of groundwater in Longkou city were mainly $HCO_3.Cl-Na.Ca$, and $HCO_3.Cl-Ca$, showing the evolution of $HCO_3-Ca$ to $HCO_3.Cl-Na$ to $Cl-Na$ from the inland to the coastline. Stable isotopes analysis with $\delta^2H$, $\delta^{18}O$ and $^{87}Sr/^{86}Sr$ indicated the main source of groundwater was atmospheric precipitation. The SI degree was the strongest at the junction of the west and north coast zones, with high values of $\delta^2H$ and $\delta^{18}O$. The high $Sr^{2+}$ concentration of groundwater was mainly from SI and groundwater–rock interactions. In the SI process, the mixing of seawater and fresh water took place first, and then different degrees of cation exchange and mineral dissolution and sedimentation occurred. Results of reverse hydrogeochemical simulation showed dolomite and quartz precipitated, with negative migrated masses of $1.38 \times 10^{-3}$ and $1.08 \times 10^{-5}$ mol/L on simulation Path 1, respectively, where calcite, halite, and gypsum dissolved with positive migrated masses of $2.89 \times 10^{-3}$, $3.52 \times 10^{-3}$, and $4.66 \times 10^{-4}$ mol/L, respectively, while dolomite and gypsum precipitated and calcite, halite, and quartz dissolved on simulation Path 2. On simulation Path 3, the dolomite, gypsum, halite, and quartz were dissolved, and calcite was precipitated, with a negative migrated mass of $1.77 \times 10^{-4}$ mol/L.

**Keywords:** seawater intrusion; Longkou city; traditional hydrogeochemistriy; stable isotope; reverse hydrogeochemical simulation



## 1. Introduction

Seawater intrusion (SI) is a universal global problem [1,2], threatening the underground freshwater quality [3] and bringing a series of ecological environmental problems, such as agricultural production reduction, soil salinization, and the development restriction of social economy. Actually, SI is a complex multi-component chemical reaction process [4], and the invasion process is not only the mixing of seawater and fresh water (SFW), but also the hydrogeochemical processes of cation exchange and mineral dissolution and

precipitation [5]. This has been paid attention to by many people. Kima et al. [6] have studied the hydrogeochemical and isotopic characteristics of the eastern and southeastern regions in Jizhou volcanic island, and confirmed the existence of cation exchange in the SI process. Mohanty et al. [7]. have showed that the ion exchange process is an important factor to control groundwater evolution, which is continuously affected by seawater mixing, carbonate mineral dissolution and water–rock interactions. Ghandour et al. [8] have analyzed the hydrochemical composition of groundwater in Egypt and Nile Delta, and determined that the east and north of the Delta were invaded by seawater and Suez Canal water. Appelo et al. [9] have proven the existence of cation exchange by using a laboratory soil column experiment of mixing SFW, and simulated the SI process by using a multi-component transport model. Werner et al. [10] have analyzed and summarized the occurrence process, monitoring technology, simulation prediction, and management of SI, and pointed out the challenges of numerical simulation for SI.

Moreover, the conventional ion-tracing method is often affected by the pollution discharge and background value. In order to overcome this deficiency, stable isotopes are used to trace SI. Additionally, the stable isotopic composition of seawater is often different from that of groundwater. When SI occurs, the isotopic characteristics of groundwater will change. Therefore, using isotopes to trace the SI process can reveal the water supply source and salinization origin, and many people have studied the SI process with stable isotopes. Chandrajith et al. [11] have quantified groundwater–seawater interaction in a coastal sandy aquifer system with stable isotopes of oxygen and hydrogen from Panama, Sri Lanka. Chandrajith et al. [12] have further used a stable isotope composition of Jaffna water to indicate the groundwater derivation in coastal karstic aquifers in northern Sri Lanka. Gemitzi et al. [13] have used stable isotopes in combination with hydrogeological and hydrochemical data to investigate the complex interaction among groundwater and a coastal lake in northwestern Greece. Louvat et al. [14] have explored the source of salinity in the Aspo groundwater system by combining the ratios of conserved ions and stable isotopes of water, and concluded that the salinity of groundwater originated from SI in the Baltic Sea. Tran et al. [15] have studied the hydrogen and oxygen stable isotope characteristics of groundwater and surface water in the coastal area of the Mekong Delta in Vietnam, and concluded that the isotopic composition of groundwater is spatially heterogeneous, reflecting different recharge sources and SI processes. Xiong et al. [16] have analyzed the hydrochemical characteristics and hydrogen and oxygen isotope characteristics of the Dagu River Basin in Qingdao, and they showed the factors affecting the formation of groundwater chemical characteristics were mainly SI.

In recent years, with rapid development of China's economy, the groundwater utilization in the Bohai Rim region has increased year by year, and the SI problem caused by large-scale groundwater exploitation has become increasingly prominent [17]. Therefore, it is necessary to study the hydrogeochemical role of groundwater in the SI process around Bohai Sea, and many scholars have done much study on the SI in Longkou city of Bohai Sea. Wu et al. [18] have summarized the development and evolution of SI in Longkou city with field observed data, analyzing the hydrochemical characteristics. Xue et al. [19] have developed a three-dimensional mathematical model to simulate the SI process in Huangheying area of Longkou city, considering the influences of wide transition zones, the fluctuation of groundwater table, and rainfall infiltration. Miao et al. [17] have established a three-dimensional variable density mathematical model of SI in Longkou city and proposed a construction scheme suitable for the study area. Zhang et al. [20] have explored the dynamic SI system by analyzing the temporal and spatial distribution of chloride concentration in groundwater of Longkou city. Li et al. [21] have described the groundwater flow field in Longkou city, and they found that the higher the groundwater level is, the smaller the migration speed of the sea-land transition zone is. Chen et al. [22] have discussed the SI impact on microbial community diversity and groundwater quality in the transition zone of SFW in Longkou city.

However, a study on stable isotope characteristics has not been investigated yet in Longkou city. Moreover, at present, the evaluation methods of SI mainly include traditional hydrogeochemistry methods, i.e., single index method, principal component, mathematical statistics method, attribute identification method. However, the calculation results of the single index method have some defects, such as uncertainty and one sidedness. The analysis results of mathematical statistics method are relatively related to the sample data, which are easy to be fluctuated, and the selection of confidence level is difficult to grasp by the attribute identification method [23]. This demands new investigation methods into the SI process and water supply sources in Longkou city.

Therefore, the purpose of this study was to study the SI in Longkou city by firstly combining the isotope method and traditional hydrogeochemistry. It should be noted that this is the first time to use isotope method to study SI in Longkou city. Additionally, the reverse hydrogeochemical simulation was also used in the study area. The diagrammatic sketch of a flowchart to study the SI in Longkou city is shown in Figure 1, where the hydrogeochemistry, stable isotope method, and reverse hydrogeochemical simulation were used sequentially.

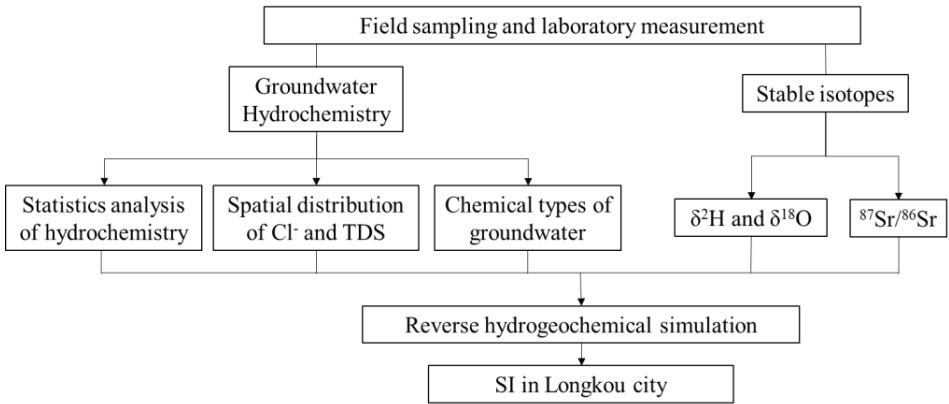

**Figure 1.** Diagrammatic sketch of flowchart to study the SI in Longkou city.

The rest of the manuscript is organized as follows. In Section 2, the geographical location, hydrogeology condition of Longkou city, and the sampling and analysis method are introduced. The hydrochemistry and stable isotopes results are provided in Sections 3 and 4. Section 5 presents the results of reverse hydrogeochemical simulation method. Major conclusions are given in Section 6.

Highlights

1.　It is the first time to use isotope method to study seawater intrusion in Longkou city.
2.　The traditional hydrogeochemical method is combined with the stable isotope method.
3.　The hydrogeochemical effect on the specific path of the study area was quantitatively analyzed.
4.　Stable isotopes analysis with $\delta^2H$, $\delta^{18}O$ and $^{87}Sr/^{86}Sr$ indicated the main source of groundwater was atmospheric precipitation.
5.　The SI degree was the strongest at the junction of the west and north coast zones, where $\delta^2H$ and $\delta^{18}O$ also had high values

## 2. Materials and Methods

### 2.1. Study Area

The location plan of Longkou city is shown in Figure 2. From Figure 2a,b, Longkou city, in Shandong province, is located in the northwest of Jiaodong Peninsula, with a longitude from 120°13′14″ to 120°44′46″ and latitude from 37°27′30″ to 37°47′24″. It is bordered by Bohai Sea in the west and north, by Qixia city and Zhaoyuan city in the south and by Penglai city in the west. Its area is 893.32 km$^2$, similar to a maple leaf in shape, with a length of 46.08 km from east to west and a width of 37.43 km from south to north, and a

coastline length of 68.38 km. The coastal zone is divided into the western and the northern coastal zones.

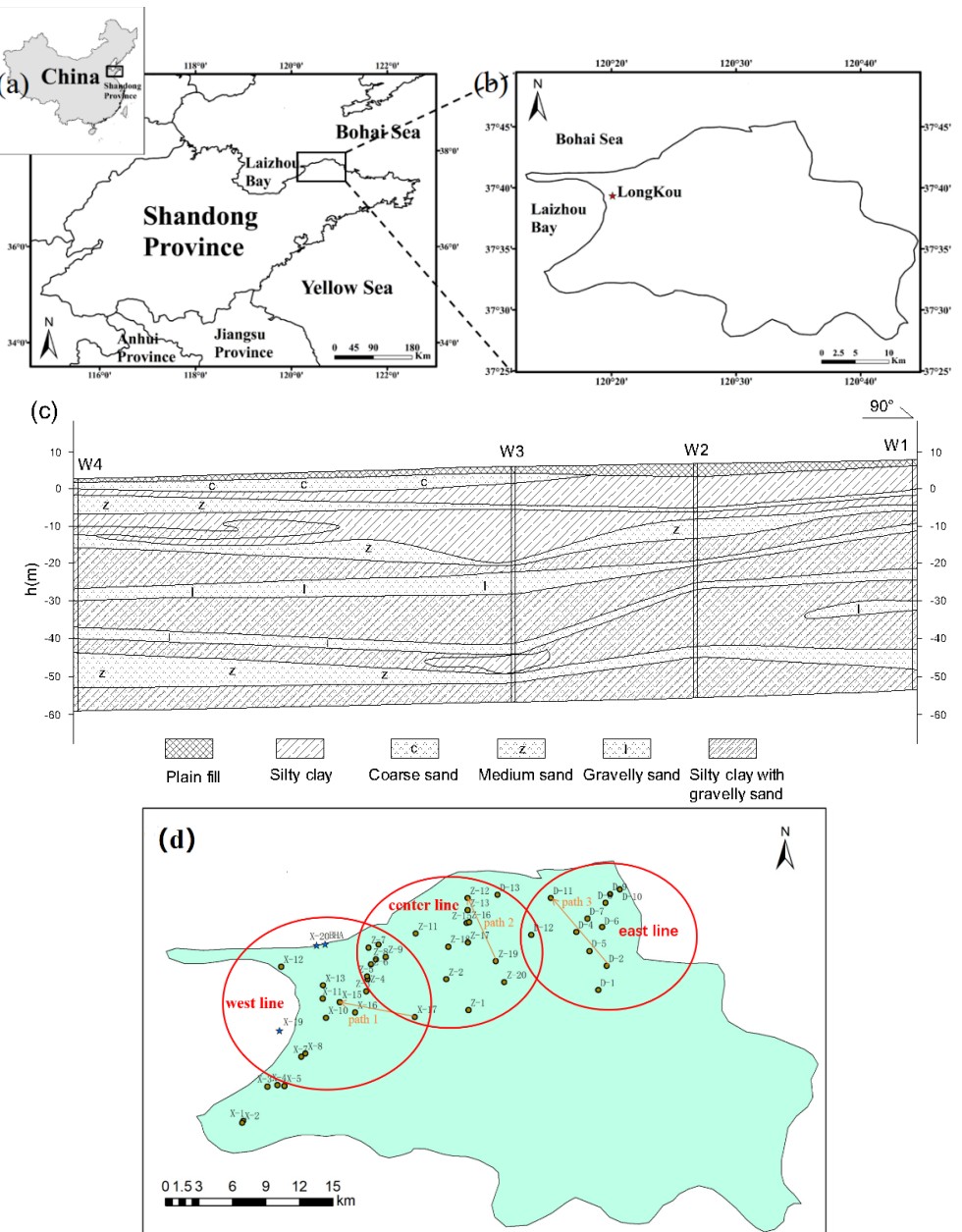

**Figure 2.** Distribution of plane view for geographical location in Longkou City and sampling points: (**a**) Plane view of Shandong Province; (**b**) Plane view of Longkou city; (**c**) Vertical aquifer lithology; (**d**) Sampling points' distribution and distribution of reverse simulation paths.

This area is a typical warm temperate semi humid monsoon continental climate with four distinct seasons. Because it is close to the Bohai Sea, it has obvious characteristics of marine climate, which is humid, hot, and rainy, without high temperature in summer. In winter, it is controlled by Mongolian high pressure and weakened by the ocean's influence, showing the characteristics of continental dry and cold climate. The annual average temperature is about 11.8 °C, with an extreme maximum temperature of 38.3 °C, measured on 5 July 1972.

The overall terrain of this area is high in the southeast and low in the northwest. It is bordered by the watershed of mountains in the adjacent counties and cities. Therefore,

all the rivers in this area originate in the eastern and southern mountainous areas and flow in a zigzag pattern to the northwest. There are 23 rivers in total, and the main rivers are Huangshui River, Yongwen River, and Balisha River, which are seasonal rivers. The outcropped strata are Cenozoic Quaternary. The Quaternary system enters the Alluvial-diluvial inclined plain from south to north through the residual slope platform. The surface layer is mostly sub-clay and sub-sandy soil, and the underlying layers are of sand and gravel. Aquifers in the area are mostly of 2–3 layers, with a total thickness of 1–15 m and an average thickness of 6.2 m. The aquifer lithology is mainly coarse sand and medium sand, followed by gravel, mostly containing a small amount of clay-filled soil, which is shown in Figure 2c. The groundwater depth is generally about 6.2 m, and the aquifer is porous phreatic water, but micro-confined water locally.

## 2.2. Sampling and Analysis

The field sampling period was from 28 February to 2 March 2019. This sampling roughly followed the principle of sequential sampling from sea to land, and the sampling scope covered three areas of the west line, the middle line, and the east line. The distribution of sampling points is shown in Figure 2d. Sample ID and chemical compositions and isotopes of groundwater samples are shown in Table 1. In Figure 2d and Table 1, samples in the study area are divided into a western line with X-N sampling points, center line with Z-N sampling points, and an eastern line with Z-N sampling points according to the region, where N is the number. From the Figure 2d and Table 1, 44 groundwater samples, including 14 groundwater samples on the west line, 18 groundwater samples on the middle line, 12 groundwater samples on the east line, and 3 seawater samples with sample ID BHA, X-19, X-20 in Table 1 were collected in the study area. Among them, groundwater were mainly collected from civil wells, which were shallow groundwater. Most wells were buried 10m below the ground's surface, and seawater was mainly collected from surface seawater. In the field, the longitude and latitude of sampling points were measured by GPS positioning. Measurements of $HCO_3^-$ were conducted by titration with a German alkalinity test box, and a German WTW Multi 3400i water quality analyzer was used to measure the temperature, conductivity, TDS, pH, and other physical indicators. Several readings were taken after equilibrium was achieved between the sondes and groundwater, and stable values were used. The sondes were checked before every use and calibration was carried out using standard solutions [24]. Groundwater was collected from civilian wells in use and seawater was collected from surface seawater. Water samples were collected in a 500 mL polyethylene bottle and were washed with 10% hydrochloric acid before sampling, and then rinsed with pure water for 2–3 times. In the field sampling site, each sample bottle was washed 2–3 times with the water sample before sampling. All samples were filtered by 0.22 um membrane. Water samples for cation analysis were acidified with high grade pure nitric acid until pH < 2, while samples used for anion and hydrogen and oxygen isotope analysis were not acidified. All samples were fully filled with sampling bottles to avoid bubble formation, and they were sealed with tape, protecting samples from light, and then were stored at normal atmospheric temperature. $Na^+$, $K^+$, $Ca^{2+}$, $Mg^{2+}$, $Cl^-$, $NO_3^-$, $SO_4^{2-}$, and other conventional ions were determined by an ICS-1600 (Thermo, USA) ion chromatograph with units of milligrams per liter in the analysis, conducted in the test center of Jinan University. Hydrogen and oxygen stable Isotopes were determined by a Liquid Water Isotope Analyzer (Picarro L2130-i) in the Groundwater and Geoscience Laboratory of Jinan University. The determination accuracy for $\delta^{18}O$ and $\delta D$ was $\pm 0.3‰$ and $\pm 1‰$, respectively. The results are expressed in the form of deviation from the Vienna standard average VSMOW. For Sr isotopic analysis, Sr was separated using a standard cation exchange technique. The Sr isotopic ratios were measured using a VG54-30 thermal ionization mass spectrometer.

**Table 1.** Chemical compositions and isotopes of groundwater samples.

| Sample ID | Ca mg·L$^{-1}$ | Mg mg·L$^{-1}$ | Na mg·L$^{-1}$ | K mg·L$^{-1}$ | HCO$_3$ mg·L$^{-1}$ | SO$_4$ mg·L$^{-1}$ | Cl mg·L$^{-1}$ | TDS mg·L$^{-1}$ | pH - | NO$_3$ mg·L$^{-1}$ | D - | $^{18}$O - |
|---|---|---|---|---|---|---|---|---|---|---|---|---|
| X-1 | 241.60 | 41.77 | 100.60 | 0.83 | 268.49 | 105.12 | 216.88 | 1873.00 | 7.33 | 222.34 | −55.49 | −7.69 |
| X-2 | 292.80 | 29.89 | 107.00 | 1.63 | 176.96 | 97.42 | 310.08 | 2020.00 | 7.72 | 427.48 | −52.99 | −7.26 |
| X-3 | 219.00 | 29.72 | 90.54 | 0.76 | 134.24 | 79.07 | 271.56 | 1473.00 | 7.52 | 367.87 | −56.67 | −7.85 |
| X-4 | 186.40 | 27.32 | 96.72 | 1.15 | 158.65 | 94.32 | 279.48 | 1398.00 | 7.48 | 149.94 | −56.96 | −7.94 |
| X-5 | 128.90 | 18.43 | 61.15 | 1.21 | 146.45 | 49.68 | 151.65 | 996.00 | 7.67 | 177.67 | −57.97 | −7.88 |
| X-7 | 188.20 | 24.23 | 74.07 | 0.54 | 280.69 | 71.33 | 224.75 | 1381.00 | 7.65 | 157.38 | −57.29 | −7.85 |
| X-8 | 187.90 | 25.00 | 66.50 | 0.69 | 268.49 | 73.62 | 213.98 | 1267.00 | 7.61 | 238.49 | −57.57 | −7.93 |
| X-10 | 234.80 | 33.50 | 193.90 | 2.42 | 292.90 | 164.87 | 522.30 | 2116.00 | 7.68 | 267.43 | −57.41 | −7.95 |
| X-11 | 265.20 | 37.92 | 158.30 | 1.26 | 360.02 | 449.90 | 354.29 | 2134.00 | 7.74 | 165.55 | −54.32 | −7.47 |
| X-12 | 920.65 | 490.85 | 4850.20 | 40.81 | 256.28 | 1653.70 | 10,465.07 | 23,230.00 | 7.20 | 0.00 | −38.90 | −4.78 |
| X-13 | 171.40 | 53.39 | 119.40 | 15.72 | 268.49 | 540.04 | 117.50 | 1513.00 | 7.84 | 83.59 | | |
| X-15 | 235.90 | 26.92 | 91.69 | 1.73 | 299.00 | 99.30 | 277.13 | 1713.00 | 7.54 | 430.31 | −56.91 | −8.05 |
| X-16 | 151.40 | 19.36 | 65.79 | 1.46 | 195.26 | 63.86 | 216.14 | 1295.00 | 7.69 | 241.14 | −59.64 | −8.05 |
| X-17 | 143.60 | 24.61 | 58.36 | 1.56 | 225.77 | 68.52 | 169.73 | 1240.00 | 7.35 | 255.06 | −58.56 | −8.05 |
| Z-1 | 73.63 | 24.22 | 56.34 | 2.02 | 280.69 | 66.64 | 75.79 | 974.00 | 7.35 | 117.98 | −56.81 | −7.66 |
| Z-2 | 150.00 | 24.00 | 55.95 | 1.25 | 244.08 | 51.02 | 145.91 | 960.00 | 7.33 | 171.44 | −57.06 | −7.95 |
| Z-3 | 239.70 | 32.11 | 77.23 | 1.47 | 207.47 | 106.90 | 231.25 | 1931.00 | 7.01 | 398.79 | −57.37 | −8.13 |
| Z-4 | 278.50 | 28.97 | 95.23 | 1.24 | 268.49 | 148.81 | 327.71 | 1871.00 | 7.15 | 270.67 | −57.09 | −8.03 |
| Z-5 | 243.40 | 29.19 | 92.46 | 1.49 | 256.28 | 124.27 | 307.69 | 1731.00 | 7.38 | 261.87 | −55.82 | −7.83 |
| Z-6 | 232.50 | 28.71 | 87.14 | 1.61 | 256.28 | 173.19 | 225.55 | 1652.00 | 7.09 | 338.55 | −56.39 | −7.98 |
| Z-7 | 111.70 | 15.12 | 110.60 | 2.22 | 299.00 | 131.33 | 184.69 | 1128.00 | 7.18 | 42.41 | −55.19 | −7.73 |
| Z-8 | 227.60 | 27.89 | 86.22 | 1.48 | 292.90 | 178.84 | 232.01 | 1646.00 | 7.07 | 349.02 | −57.17 | −7.77 |
| Z-9 | 227.30 | 32.03 | 150.70 | 2.11 | 463.75 | 284.50 | 264.77 | 1944.00 | 7.34 | 287.64 | −53.46 | −7.19 |
| Z-10 | 167.30 | 22.45 | 53.57 | 1.34 | 378.32 | 179.54 | 115.66 | 1050.00 | 7.3 | 86.25 | −53.60 | −7.47 |
| Z-12 | 142.70 | 42.32 | 90.44 | 1.97 | 317.30 | 72.53 | 289.95 | 1515.00 | 7.39 | 175.18 | −58.37 | −8.03 |
| Z-13 | 209.20 | 46.22 | 107.20 | 1.52 | 396.63 | 126.53 | 392.61 | 1953.00 | 7.26 | 280.69 | −55.88 | −7.64 |
| Z-15 | 262.50 | 37.80 | 127.30 | 1.32 | 488.16 | 131.86 | 377.75 | 2136.00 | 7.21 | 443.17 | −55.56 | −7.72 |
| Z-16 | 238.10 | 34.84 | 109.80 | 1.22 | 360.02 | 156.77 | 317.14 | 1775.00 | 7.21 | 432.21 | −59.64 | −8.02 |
| Z-17 | 246.90 | 47.51 | 85.91 | 1.80 | 396.63 | 123.32 | 230.89 | 2046.00 | 7.27 | 636.77 | −59.75 | −8.18 |
| Z-18 | 127.90 | 20.41 | 175.00 | 1.63 | 506.47 | 95.19 | 313.72 | 1628.00 | 7.47 | 67.40 | −58.04 | −8.01 |
| Z-19 | 92.32 | 27.80 | 45.65 | 7.42 | 146.45 | 316.08 | 57.27 | 858 | 7.67 | 41.504 | | |
| Z-20 | 93.63 | 28.53 | 46.20 | 7.49 | 183.06 | 316.59 | 58.97 | 889 | 7.67 | 43.038 | | |
| D-1 | 68.95 | 16.91 | 46.69 | 1.10 | 445.45 | 78.53 | 46.40 | 944 | 7.52 | 83.716 | −53.22 | −7.55 |
| D-2 | 80.21 | 25.94 | 65.12 | 2.55 | 176.96 | 170.71 | 83.50 | 954 | 7.36 | 121.537 | −52.27 | −7.02 |
| D-4 | 183.00 | 39.07 | 191.90 | 3.81 | 262.39 | 246.38 | 502.81 | 2051 | 7.23 | 106.59 | −53.73 | −6.90 |
| D-5 | 143.80 | 39.62 | 77.02 | 3.87 | 274.59 | 236.65 | 142.71 | 1378 | 7.45 | 208.405 | −50.78 | −6.50 |
| D-6 | 181.60 | 28.65 | 94.07 | 2.64 | 433.24 | 182.16 | 154.41 | 1452 | 7.39 | 182.935 | −54.57 | −7.52 |
| D-7 | 297.40 | 50.78 | 174.90 | 3.45 | 439.34 | 382.12 | 468.59 | 2609 | 7.36 | 394.746 | −55.41 | −7.50 |
| D-8 | 207.90 | 34.53 | 99.43 | 1.93 | 335.61 | 131.71 | 370.36 | 1674 | 7.25 | 156.374 | −58.07 | −8.03 |
| D-9 | 114.40 | 21.10 | 46.47 | 2.38 | 237.98 | 117.15 | 111.73 | 938 | 7.44 | 114.444 | −57.04 | −7.78 |
| D-10 | 171.10 | 48.23 | 80.18 | 3.46 | 378.32 | 205.53 | 221.21 | 1578 | 7.49 | 243.357 | −56.76 | −7.68 |
| D-11 | 200.50 | 44.75 | 129.80 | 4.60 | 268.49 | 299.78 | 379.57 | 2013 | 6.99 | 253.455 | −56.73 | −7.45 |
| D-12 | 312.50 | 60.96 | 63.06 | 4.57 | 207.47 | 204.38 | 377.91 | 2510 | 7.54 | 828.193 | −58.31 | −7.85 |
| D-13 | 152.30 | 35.31 | 59.94 | 2.19 | 317.30 | 120.36 | 219.65 | 1303 | 7.42 | 175.438 | −57.55 | −7.80 |
| BHA | 364.40 | 1044.00 | 9814.00 | 300.00 | 176.96 | 2526.18 | 17,587.48 | 35,620 | 7.78 | 0 | −1.51 | −14.47 |
| X-19 | 392.90 | 989.20 | 9079.00 | 270.00 | 237.98 | 2400.33 | 16,437.51 | 34,030 | 7.82 | 0 | −1.12 | −10.95 |
| X-20 | 374.00 | 1056.00 | 9745.00 | 301.60 | 183.06 | 2600.47 | 17,868.81 | 38,360 | 7.69 | 0 | −1.09 | −10.82 |

Note: BHA, X-19 and X-20 were seawater.

In the following sections below, methods of correlation analysis, principal component analysis, Piper triple line map, and seawater ratio line, to analyze the characteristics and hydrochemical types of groundwater in the study area, were used, and the SI and salt source of groundwater in this area were explored. By analyzing the characteristics for strontium, hydrogen, and oxygen stable isotopes of groundwater, the recharge source of groundwater and the mixing degree of SFW in the study area were discussed. The reverse hydrogeochemical simulation technology was used to quantitatively analyze the geochemical effect on the SI process.

## 3. Hydrochemistry

### 3.1. Statistics Analysis of Hydrochemistry

Ion correlation analysis can explain the similarity and difference of hydrochemical parameters for groundwater and their corresponding sources, while TDS value represents the content of dissolved matter in groundwater, which can reveal the groundwater evolution. In this study, SPSS19.0 software was used for correlation analysis of Ca$^{2+}$, Mg$^{2+}$, Na$^+$,

$K^+$, $HCO_3^-$, $SO_4^{2-}$, $Cl^-$, $NO_3^-$, TDS, and pH, and the results are shown in Table 2. From Table 2, it can be seen that the TDS of groundwater in the Longkou city had significant correlation with main cations, and the correlation coefficient values for cations from large to small were $Na^+$, $Mg^{2+}$, $K^+$, and $Ca^{2+}$, with 0.996, 0.995, 0.958, and 0.612, respectively. TDS had high correlation with $Cl^-$ and $SO_4^{2-}$ with values of 0.999 and 0.987, while it had significant negative correlation with $HCO_3^-$ as value of $-0.231$. In addition, the correlations between TDS and $Cl^-$ and $Na^+$ were the most significant, suggesting that $Cl^-$ and $Na^+$ contributed the most to TDS in coastal areas. The high correlation coefficients of $Na^+$, $Mg^{2+}$, $K^+$, and $Cl^-$ were 0.998, 0.996, and 0.964, respectively, which suggested that they had similar sources, mainly from SI. Meanwhile, $K^+$, $Na^+$, $Mg^{2+}$, and $SO_4^{2-}$ were significantly correlated with coefficients of 0.950, 0.984, and 0.984, respectively, indicating that their sources were similar, mainly from the dissolution of sulfate. The correlation coefficients of $NO_3^-$ with TDS, $Na^+$, $Mg^{2+}$, $K^+$, $SO_4^{2-}$, and $Cl^-$ were negative values of $-0.373$, $-0.396$, $-0.376$, $-0.372$, $-0.413$, and $-0.392$, respectively, and $NO_3^-$ content in groundwater was low, which indicated that the study area was less affected by industrial and agricultural activities.

**Table 2.** Pearson correlation coefficient of common ions and hydrogen and oxygen isotopes.

| | pH | TDS | $K^+$ | $Na^+$ | $Ca^{2+}$ | $Mg^{2+}$ | $HCO_3^-$ | $Cl^-$ | $SO_4^{2-}$ | $NO_3^-$ | $\delta D$ | $\delta^{18}O$ |
|---|---|---|---|---|---|---|---|---|---|---|---|---|
| pH | 1 | | | | | | | | | | | |
| TDS | −0.514 | 1 | | | | | | | | | | |
| $K^+$ | −0.562 | 0.958 ** | 1 | | | | | | | | | |
| $Na^+$ | −0.547 | 0.996 ** | 0.977 ** | 1 | | | | | | | | |
| $Ca^{2+}$ | −0.203 | 0.612 ** | 0.385 ** | 0.550 ** | 1 | | | | | | | |
| $Mg^{2+}$ | −0.542 | 0.995 ** | 0.981 ** | 0.999 ** | 0.540 ** | 1 | | | | | | |
| $HCO_3^-$ | 0.122 | −0.231 | −0.252 | −0.242 | −0.054 | −0.241 | 1 | | | | | |
| $Cl^-$ | −0.533 | 0.999 ** | 0.964 ** | 0.998 ** | 0.591 ** | 0.996 ** | −0.241 | 1 | | | | |
| $SO_4^{2-}$ | −0.511 | 0.987 ** | 0.950 ** | 0.984 ** | 0.599 ** | 0.984 ** | −0.222 | 0.986 ** | 1 | | | |
| $NO_3^-$ | 0.167 | −0.373 | −0.372 | −0.396 | 0.035 * | −0.376 | 0.137 | −0.392 | −0.413 | 1 | | |
| $\delta D$ | −0.101 | 0.906 ** | 0.811 ** | 0.803 ** | 0.670 ** | 0.788 ** | 0.017 | 0.910 ** | 0.835 ** | −0.380 | 1 | |
| $\delta^{18}O$ | −0.097 | 0.868 ** | 0.830 ** | 0.798 ** | 0.656 ** | 0.797 ** | −0.012 | 0.874 ** | 0.851 ** | −0.342 | 0.919 ** | 1 |

Note: ** Coefficients at the 0.01 significance level, $p < 0.01$. * Coefficients at the 0.05 significance level, $p < 0.05$.

Using the factor analysis module of SPSS19.0 software, the maximum variance rotation was used to analyze the nine chemical—TDS, $Mg^{2+}$, $Cl^-$, $Na^+$, $SO_4^{2-}$, $K^+$, $Ca^{2+}$, $NO_3^-$, $HCO_3^-$—indexes, with units of milligrams per liter for the groundwater in Longkou city. Based on the cumulative variance contribution rate of greater than 85% and each principal component eigenvalue greater than one two principal components were extracted finally. Element principal components of load matrix are also shown in Figure 3. According to Figure 3a, the eigenvalues of principal components one and two were 6.577 and 1.159, respectively, and the corresponding variance contribution rates were 73.081 and 12.883%, respectively. In addition, the cumulative contribution rates of principal components one and two were 73.081 and 85.964%, respectively. Because the variance contribution rate of principal component one (73.081%) was much higher than that of principal component two (12.883%), the hydrochemical genesis of groundwater in the study area was mainly determined by principal component one.

From Figure 3b, it can be seen that main component one was mainly TDS, $Mg^{2+}$, $Cl^-$, $Na^+$, $SO_4^{2-}$, $K^+$, and $Ca^{2+}$, with load values of 0.994, 0.993, 0.986, 0.985, 0.942, 0.929, and 0.926, respectively. From the Table 2, we can also find that the correlation coefficients of $Mg^{2+}$, $Na^+$, $SO_4^{2-}$, $K^+$, $Ca^{2+}$, $Cl^-$, and TDS were high, which proved that they were very similar in genesis. Therefore, it can be determined that the principal component one can represent the indicator of the SI degree. The higher the concentrations of $Mg^{2+}$, $Na^+$, $SO_4^{2-}$, $K^+$, $Ca^{2+}$, $Cl^-$, and TDS, the stronger the SI. The principal component two was mainly composed of $NO_3^-$ and $HCO_3^-$, with load matrix values of 0.889 and 0.483, respectively, indicating $HCO_3^-$ mainly came from carbonate rocks and $NO_3^-$ mainly represented

the pollution caused by industrial and agricultural activities. Therefore, the principal component two represented the degree of carbonate dissolution and human pollution.

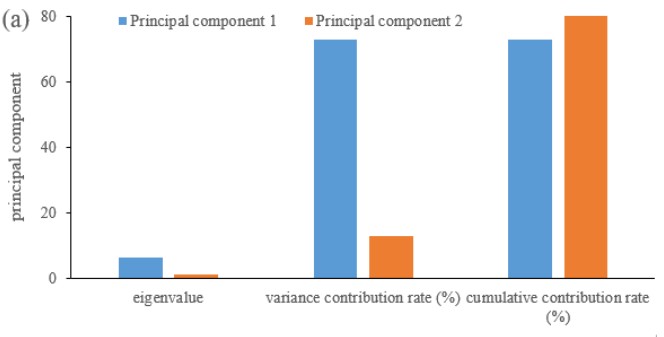

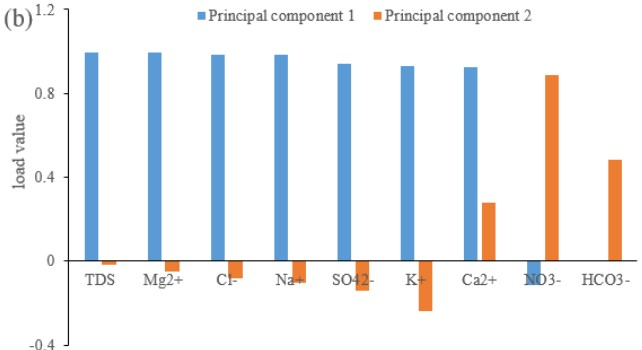

**Figure 3.** Element principal component and load matrix: (**a**) principal component; (**b**) load matrix.

### 3.2. Spatial Distribution of Cl⁻ and TDS

$Cl^-$ and TDS were considered as the main component one in this study, which are the most commonly used indicators for judging SI [22]. Therefore, the spatial distribution of $Cl^-$ and TDS are shown in Figure 4 to explore the SI, where the Kriging method was used for interpolation. The Pearson correlation coefficients of common ions are shown in Table 2. From Table 2, we can see that the correlation coefficient between $Cl^-$ and TDS was 0.999, which further indicated that $Cl^-$ occupied a large proportion and was the main source of TDS, so its variation trend was basically the same. It can be seen from the Figure 4 that the farther away from the coastline one is, the lower the concentrations of $Cl^-$ and TDS are, and the higher the concentrations of $Cl^-$ and TDS are near the coastal zone. The $Cl^-$ and TDS concentrations at the junction of the west and north coast zones were very high, due to the combined action of the two coastal zones, where the SI degree was the strongest. The $Cl^-$ and TDS concentrations close to the eastern coastline were higher, which may be the result of more human activities near the Huangshui River, and the combined effects of industrial wastewater and agricultural sewage pollution, as well as seawater, leading to the increasing concentration of $Cl^-$ and TDS in groundwater. From the overall trend, the ion exchange function was dominant in the west coast. Because of the long-term overexploitation of groundwater in the west coast, such as the large amount of groundwater use in power stations, a serious landing subsidence appeared. So, there was a large hydraulic gradient, which made chloride-rich seawater enter the west coast. Due to the leaching erosion of seawater in the coastal zone, the pore structure of groundwater media was changed. The closer to the coastal zone, the greater the porosity, and the looser the soil structure, which further promoted the ion exchange. It is shown from Figure 4 that the closer one is to the coastal zone, the more obvious the ion exchange function is. Meanwhile, the TDS concentration in the west coast was high, but the $Cl^-$ concentration was relatively low, which indicated that other ions in this area contributed much more than $Cl^-$ to TDS.

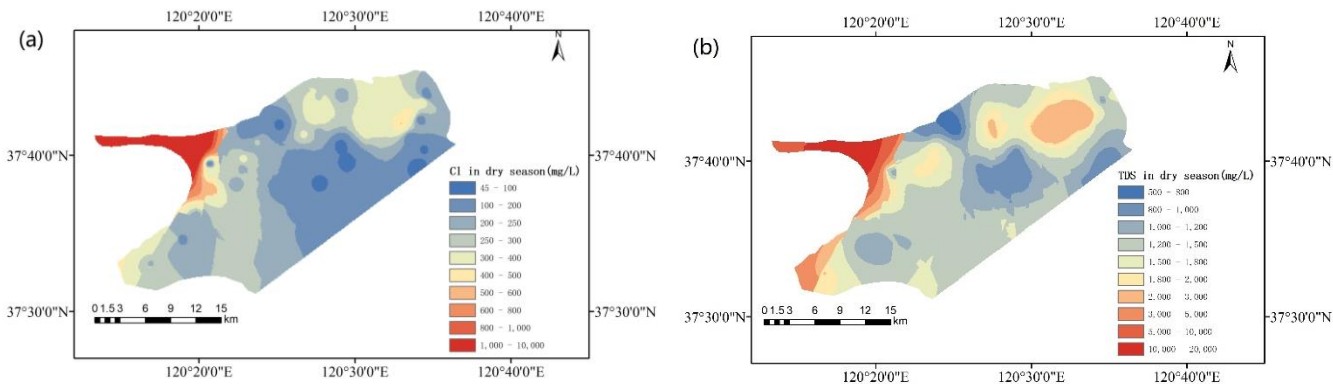

**Figure 4.** The spatial distribution of Cl$^-$ and TDS: (**a**) Cl$^-$; (**b**) TDS.

### 3.3. Chemical Types and Salt Source of Groundwater

Since analyzing hydrochemical types of groundwater has a significance to reveal the spatial distribution and hydrogeochemical composition in groundwater [25], chemical types and salt source were analyzed in the study area based on the above analysis. As shown in Figure 2d, groundwater samples in the study area were divided into western line groundwater, center line groundwater, and eastern line groundwater. Groundwater samples in each line were divided into fresh water, brackish water, and brine, based on the TDS concentration. Freshwater refers to water with TDS less than 1 g/L, brackish water refers to water with TDS greater than 1 g/L and less than 3 g/L, salt water refers to water with TDS greater than 3 g/L and less than 50 g/L, and brine refers to water with TDS greater than 50 g/L. Different types of groundwater samples in the study area were projected onto the piper triple line map, as shown in Figure 5. In the figure, the brackish water was in the three lines, while the fresh groundwater were in the east and middle lines. Furthermore, it can be seen from Figure 5 that the chemical types of groundwater were mainly HCO$_3$·Cl-Ca·Mg and HCO$_3$·Ca·Mg, and their distributions were concentrated on the piper triple line map. This was because the study area was affected by different degrees of cationic alternating adsorption, which made the concentrations of Mg$^{2+}$ and Ca$^{2+}$ higher. In addition, the Cl$^-$ concentration in groundwater samples closer to the coastline was higher, which was lower farther away from the coastline. Meanwhile, X-19 and X-20 on the western line were seawater, which belonged to the typical Na–Cl type water, due to the high concentration of Cl$^-$ and Na$^+$ in seawater. X-12 on the western line was located in the lower right part of the piper triple line map in Figure 5, which was brine. Further analysis showed that X-12 was the closest sample point to the coastline, where the Cl$^-$ and TDS concentrations were similar to that of seawater, indicating that it was also a typical Na–Cl type water affected by strong SI.

In order to further explore the mixing degree of groundwater and seawater and the dissolution of rocks and minerals in the study area, the ratio of each standard seawater ion concentration to Cl$^-$ concentration, named the seawater ratio was used in this study. The relationship between the main ions ratios of Ca$^{2+}$, Mg$^{2+}$, Na$^+$, K$^+$, SO$_4$$^{2-}$, and HCO$_3$$^-$ to Cl$^-$ and Cl$^-$ concentration are shown in Figure 6 in terms of logarithmic coordinates, which were compared with those of seawater ratio to reveal the mechanism of groundwater salinization. The horizontal line TSDL in the figure was the standard seawater ratio line of the corresponding ions. The ratio of the main ions to Cl$^-$ on the line is equal to the seawater ratio, so it is also called the seawater ratio line [26]. From Figure 6a, it can be seen that the ratio of HCO$_3$$^-$/Cl$^-$ in groundwater was much larger than that of standard seawater dilution line (except X-12)., while the ratio of HCO$_3$$^-$/Cl$^-$ in seawater was slightly larger than that of the seawater dilution line. The possible causes of this phenomenon were because when groundwater flowed in the aquifer, carbonate minerals in the stratum were dissolved, which increased the HCO$_3$$^-$ concentration in groundwater and resulted in a larger ratio of HCO$_3$$^-$/Cl$^-$ in the aquifer. It also may be because vegetation was more

abundant in the inland plain area than that in the coastal area, and the vegetation respiration would produce a large amount of $CO_2$. When atmospheric precipitation infiltrated into the soil zone, the $CO_2$ released by plant roots dissolved in water to form carbonate, which dissociated to form $HCO_3^-$ and $H^+$. Additionally, $H^+$ dissolved the weathered minerals in the soil zone, which made $Ca^{2+}$, $Mg^{2+}$, and other minerals components enter the groundwater with the form of $Ca^{2+}$ and $Mg^{2+}$. At the same time, several seawater points on $HCO_3^-/Cl^-$ in the figure were slightly higher than the seawater dilution line. This was because the seawater was from the surface seawater nearest to land, which may have been affected by the mixing of surface water or groundwater. In addition, the $HCO_3^-/Cl^-$ values of all samples decreased and approached the seawater dilution line with the increase of $Cl^-$ concentration. In Figure 6b,c, we can see that the ratios of $Ca^{2+}/Cl^-$ and $Mg^{2+}/Cl^-$ in groundwater samples were basically above the seawater dilution line. The possible reasons for the higher ratios of $Ca^{2+}/Cl^-$ and $Mg^{2+}/Cl^-$ were that $Ca^{2+}$ and $Mg^{2+}$ were abundant in stratum minerals. Dissolution caused $Ca^{2+}$, $Mg^{2+}$, and other components to enter groundwater, which increased the $Ca^{2+}$ and $Mg^{2+}$ concentrations in groundwater. In Figure 6d,e, we can see that the $Na^+/Cl^-$, $K^+/Cl^-$ ratios of most groundwater in the western line were lower than the standard seawater dilution line. This was because the sampling points of groundwater in the western line were all along the west coast, where the precipitation components can be regarded as strongly diluted seawater without effects of other conditions, and it should be located on the seawater dilution line theoretically. However, the sampling points of groundwater in the western line showed obvious fluctuations below the seawater dilution line, which may be due to the serious SI in the western line area. During the SI process, cationic alternating adsorption took place. The ratios of $Na^+/Cl^-$, $K^+/Cl^-$ in the mid-line groundwater, east-line groundwater, river water, and seawater all fluctuated above and below the standard seawater dilution line in Figure 6d,e. This was mainly because $Na^+$ and $K^+$ are considered to be conservative chemical elements that hardly accumulate and absorb in groundwater. It can be seen from Figure 6f that the ratio of $SO_4^{2-}/Cl^-$ in groundwater samples was basically higher than that of the seawater dilution line, while the seawater samples were generally located on the seawater dilution line. The possible reasons for the higher $SO_4^{2-}/Cl^-$ in groundwater were that the coastal area in Shandong province has had rapid economic development, which has led to a large population density and serious pollution caused by human activities. Under the effect of atmospheric rainfall, where various pollutants enter the groundwater, this would also increase the $SO_4^{2-}$ concentration.

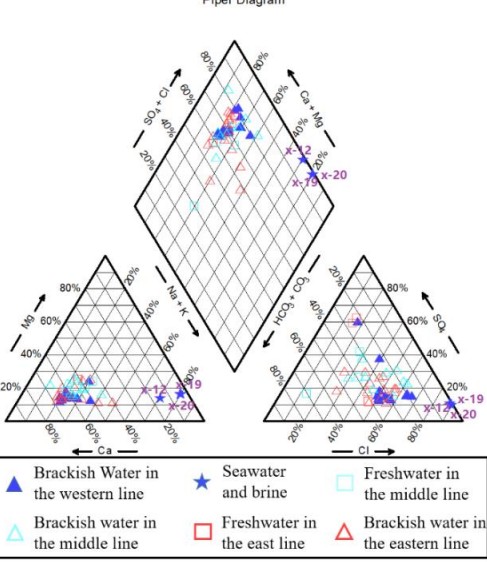

**Figure 5.** Piper triple line map of groundwater hydrochemistry.

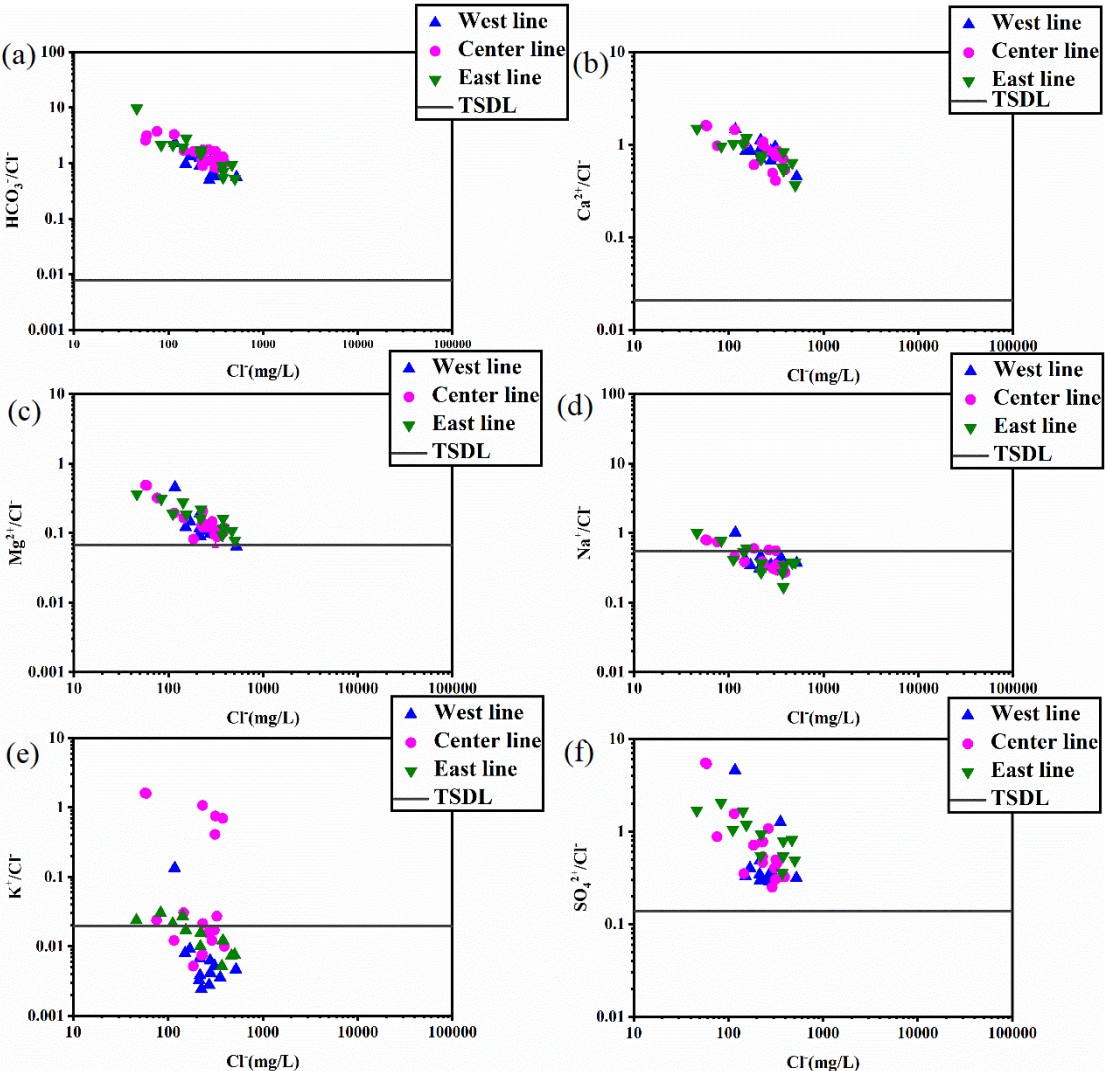

**Figure 6.** The relationship between main ions ratios to $Cl^-$ and $Cl^-$: (**a**) $HCO_3^-/Cl^-$ and $Cl^-$; (**b**) $Ca^{2+}/Cl^-$ and $Cl^-$; (**c**) $Mg^{2+}/Cl^-$ and $Cl^-$; (**d**) $Na^+/Cl^-$ and $Cl^-$; (**e**) $K^+/Cl^-$ and $Cl^-$; (**f**) $SO_4^{2-}/Cl^-$ and $Cl^-$. Note: TSDL was the standard seawater.

## 4. Stable Isotopes

### 4.1. Oxygen and Hydrogen Isotopes

Stable isotopes of oxygen and hydrogen were applied to further explore the effect of groundwater by the seawater in Longkou city. The inverse distance method of Arcgis software was used to interpolate spatial distribution of $\delta^2H$ and $\delta^{18}O$, which is shown in Figure 7. It can be seen from the Figure that both $\delta^2H$ and $\delta^{18}O$ at the junction of the west and north coast had high values. When compared with other areas, this area was rich in heavy isotopes, indicating that enrichment can be attributed to the influence of seawater. It can be seen from Table 2 that $\delta^2H$ and $\delta^{18}O$ had strong correlation with $Cl^-$ and TDS, suggesting the enrichment of $\delta^2H$ and $\delta^{18}O$ was obviously related to the increase of $Cl^-$ and TDS, so the groundwater in Longkou city was affected by the mixing of seawater. The values of hydrogen and oxygen isotopes in the D-2 sample point on the east line were relatively heavy, but the values of $Cl^-$ and TDS there were relatively low, which belonged to the typical fresh groundwater and typical inland area. This suggested the enrichment of heavy isotopes in the groundwater was not caused by SI, but by strong evaporation.

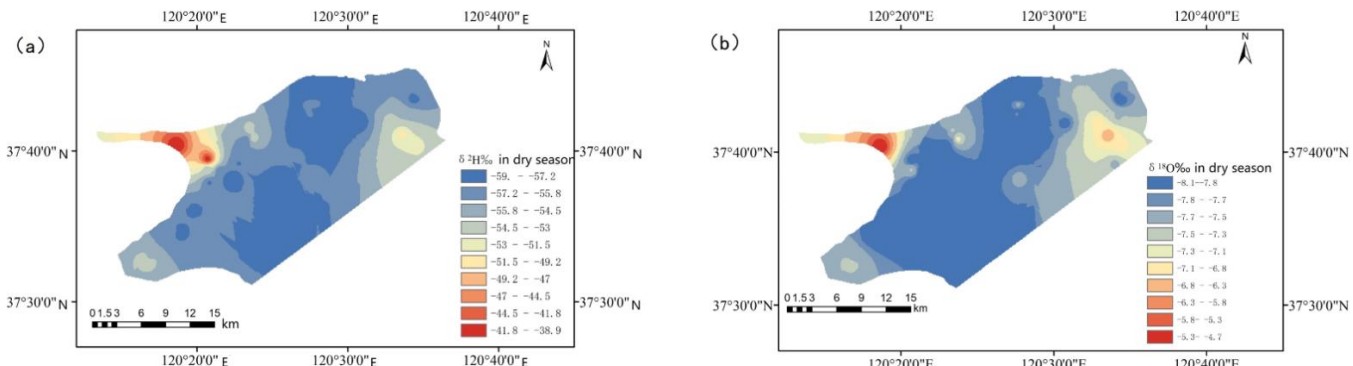

**Figure 7.** The spatial distribution of $^2$H and $^{18}$O: (**a**) $^2$H; (**b**) $^{18}$O.

The groundwater was divided into fresh water, brackish water, and salt brine. Moreover, the $\delta^2$H-$\delta^{18}$O relationship between groundwater and seawater samples on the west line, middle line, and east line are shown in Figure 8. The line $\delta^2$H = 8$\delta^{18}$O + 10 in the Figure represented the global meteoric water line (GMWL), and the line $\delta^2$H = 7.8 $\delta^{18}$O + 6.3 referred to the local atmospheric drawdown line in Yantai (LMWL). The theoretical values of $\delta^2$H and $\delta^{18}$O for the standard seawater are both zero, because of the unchanged isotopic composition and physicochemical properties of the seawater as a whole, and because the distribution characteristics of the isotopes have the trend to homogenize. The relationship between $\delta^2$H and $\delta^{18}$O on the west line is shown in Figure 8a. In the Figure, X-19, X-20, and BHA were seawater, X-12 was salt brine, X-5 was fresh water, and the rest were brackish water. Most groundwater samples were concentrated at the lower right part of the GMWL, which is shown in Figure 8b to see them clearly. The average values of $\delta^2$H and $\delta^{18}$O at the sea point on the west line in this study was −1.24 and −12.08, respectively. It can be seen from the Figure that the distribution of seawater obviously deviated from the GMWL and approached the standard seawater point, but the values of $\delta^2$H and $\delta^{18}$O were lower than those of the standard seawater point. This was because the seawater taken at this time was the surface seawater, and the hydrogen and oxygen isotopic composition of the surface seawater changed greatly. Due to the influence of the land water, its isotopic composition was often negative. The variation ranges of $\delta^2$H and $\delta^{18}$O composition for groundwater were from −59.64 to −38.90‰ and from −8.05 to −4.78‰, with averages of −55.44 and −7.59‰, respectively. Among them, the characteristics of hydrogen and oxygen stable isotope composition for fresh groundwater and brackish water were similar, which were all distributed near the GMWL. And they fell on the right lower part of the GMWL, indicating that they mainly came from atmospheric precipitation and were affected by evaporation and the SI. The location of brine water at X-12 on the west line in the figure was far away from the GMWL, suggesting that there were other important supply sources besides atmospheric precipitation. Further analysis showed that X-12 was the nearest groundwater sample point to the coastal zone, so its hydrogen and oxygen isotope values were close to the seawater, while the Cl$^-$ and TDS concentrations close to the seawater values at point X-12 indicated it was affected by strong seawater mixing.

The $\delta^2$H-$\delta^{18}$O relationship of groundwater on the middle line is shown in Figure 8c, and groundwater was divided into fresh and brackish water. It can be seen from the Figure that the groundwater on the middle line was also near the GMWL, indicating that atmospheric precipitation was the main source. The variation ranges of $\delta^2$H and $\delta^{18}$O composition for fresh groundwater were from −60.82 to −56.81‰ and from −8.14 to −7.66‰, with averages of −58.64 and −8.08‰, respectively. Meanwhile, the variation ranges of $\delta^2$H and $\delta^{18}$O compositions for brackish groundwater were from −59.75 to −53.46 ‰ and from −8.18 to −7.19‰, with averages of −56.67 and −7.84‰, respectively. The reasons were similar to that on the west line in Figure 8a, so they are not described here again.

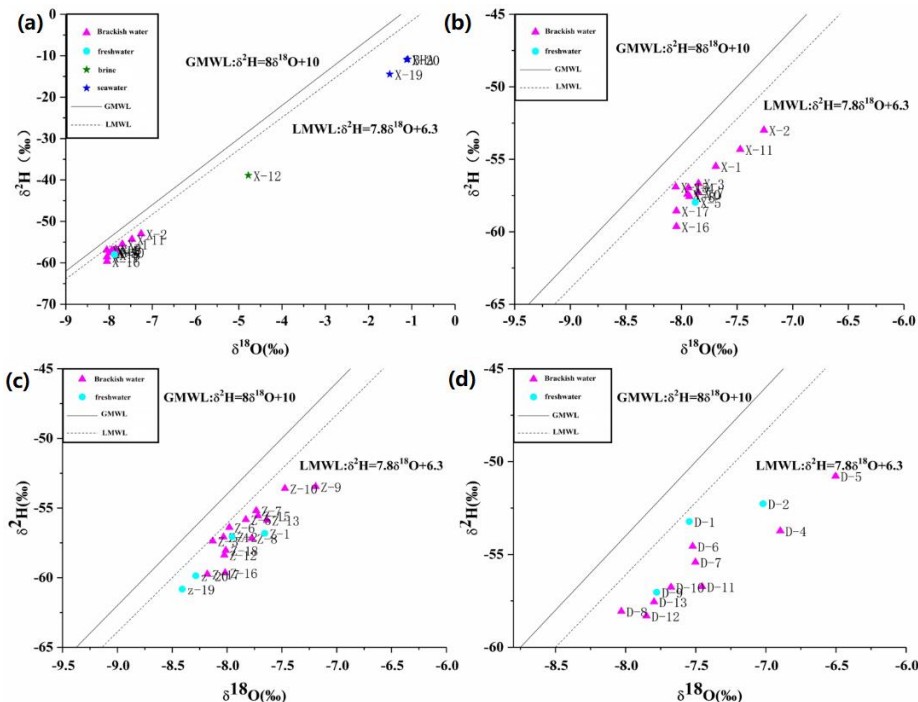

**Figure 8.** δ2H-δ18O relation of west line, center line and east line: (**a**) west line; (**b**) part of west line; (**c**) center line; (**d**) east line.

Additionally, the $\delta^2$H-$\delta^{18}$O relationship of groundwater on the east line is shown in Figure 8d. Similarly, groundwater was divided into fresh and brackish groundwater. It can be seen from the figure that the groundwater samples on the east line fell in the lower right part of the GMWL, so its main source was also atmospheric precipitation. The variation ranges of $\delta^2$H and $\delta^{18}$O compositions for the fresh groundwater were from −57.04 to −52.27‰ and from −7.78 to −7.02‰, with averages of −54.18 and −7.45‰, respectively. The variation ranges of $\delta^2$H and $\delta^{18}$O compositions for brackish groundwater were from −58.31 to −50.78‰ and from −8.03 to −6.05‰, with averages of −55.77 and −7.47‰, respectively. The characteristics of $\delta^2$H and $\delta^{18}$O compositions for fresh water and brackish water were similar. Among them, D-4 and D-5 deviated farthest from the GMWL, and the values of $\delta^2$H and $\delta^{18}$O were the highest. Further analysis showed that D-4 and D-5 were far away from the coastline and belonged to typical inland water, but the Cl⁻ and TDS concentrations were high, which indicated that the isotopic enrichment was mainly due to strong evaporation.

### 4.2. Strontium Isotopes

Groundwater sources can be obtained by the stable isotopes of $\delta^2$H and $\delta^{18}$O, and stable isotope of strontium can be further used to explore the interaction between the groundwater and aquifer, which was applied in this study. Additionally the relationships between $Sr^{2+}$ and $Ca^{2+}$ concentrations and $Sr^{2+}$ and $^{87}Sr/^{86}Sr$ in groundwater are shown in Figure 9. In Figure 9a, the $Ca^{2+}$ concentration was from 68.95 to 920.65 mg/L, where the maximum value was more than 10 times of the minimum value, with the average of 214.31 mg/L. This indicated the $Ca^{2+}$ concentration in groundwater of Longkou city varied greatly. The $Sr^{2+}$ concentration was low, between 0.926 and 6.2 mg/L, where the maximum value was more than six times of the minimum value, with an average of 1.68 mg/L, which was far higher than the $Sr^{2+}$ concentration in atmospheric precipitation. This suggested that atmospheric precipitation was not the main source of $Sr^{2+}$ in groundwater. Except for X-12, the difference of $Sr^{2+}$ concentration in other groundwater samples was very small, which was in a relatively stable range. Additionally, the $Sr^{2+}$ concentration in groundwater increased with the increase of $Ca^{2+}$ concentration. This was because $Sr^{2+}$ and

$Ca^{2+}$ are in the same main group of the Periodic Table of Elements, and they had similar hydrogeochemical characteristics, which also provided convenience to further study the groundwater–rock interaction.

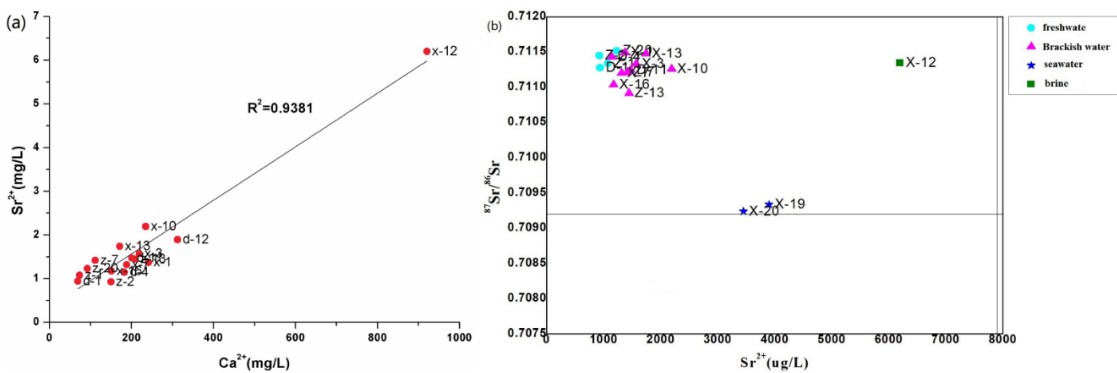

**Figure 9.** Relationship between $Sr^{2+}$ and $Ca^{2+}$ concentration and $^{87}Sr/^{86}Sr$ in groundwater: (**a**) $Sr^{2+}$ and $Ca^{2+}$; (**b**) $Sr^{2+}$ and $^{87}Sr/^{86}Sr$.

The relationship between $Sr^{2+}$ and $^{87}Sr/^{86}Sr$ is shown in Figure 9b. It can be seen from the figure that the variation ranges of $^{87}Sr/^{86}Sr$ ratios were 0.7112787–0.7115156 and 0.7109149–0.7114948 for fresh and brackish groundwater, with averages of 0.711395 and 0.711261, respectively. The $^{87}Sr/^{86}Sr$ ratios of fresh and brackish groundwater were higher than that of standard seawater, with 0.7092. It can be inferred that the main source of dissolved $Sr^{2+}$ in these groundwater samples was not from SI. The exposed strata in Longkou city are Quaternary Systems of the Cenozoic era. The weathering of soil was an important source of $Sr^{2+}$, which was reflected by the ratio of $^{87}Sr/^{86}Sr$ in groundwater. Further analysis showed that the ratios of $^{87}Sr/^{86}Sr$ in these groundwater samples were greater than 0.710, which was in the silicate weathering range, indicating the recharge of these groundwater samples was likely to be controlled by silicate. In addition, agriculture and other human activities may also be the potential reason for the high $^{87}Sr/^{86}Sr$ ratios. The variation range of $^{87}Sr/^{86}Sr$ ratio for agricultural fertilizer was 0.70791–0.71155 [27]. Additionally, the $^{87}Sr/^{86}Sr$ ratio of these groundwater samples was within this range, but it was difficult to make further evaluation because the specific fertilization type in the study area was unknown. Although the $^{87}Sr/^{86}Sr$ ratio for X-12 sample was relatively high, its $Sr^{2+}$ concentration was very high, which was close to the $Sr^{2+}$ concentration of standard seawater with 7 mg/L, indicating that the SI resulted in a high $Sr^{2+}$ concentration and the $Sr^{2+}$ of groundwater in X-12 was mainly from SI and groundwater–rock interactions. The $^{87}Sr/^{86}Sr$ ratios of seawater samples in X-19 and X-20 were close to that of standard seawater, but the $Sr^{2+}$ concentrations were far lower than the latter, which may be affected by the mixing of land water.

## 5. Reverse Hydrogeochemical Simulation

Reverse hydrogeochemical simulation is to simulate the water rock reaction on the reaction path by knowing the hydrochemical and isotopic indexes of the starting point samples on the same flow path. Therefore, reverse hydrogeochemical simulation can simulate the hydrogeochemical process on the specific path, which was further used in this study. The reverse hydrogeochemical simulation path should follow the principle that the water samples at the starting and ending points are selected from the upstream to the downstream and located in the same flow path [28]. In this study, three paths were selected for simulation along the groundwater flow direction, which is shown in Figure 2d. Path 1 was on west line with sampling points of X17-X15, Path 2 was on the middle line, with sampling points of Z19-Z12, and Path 3 was on the east line, with sampling points of D2-D11, as seen in Figure 2d. Additionally, the groundwater sample chemical compositions of these three paths are shown in Table 3. From the table, all the pH values on the three simulation

paths showed a decreasing trend and all $HCO_3^-$ concentrations showed an increasing trend. This was because $CO_2$ dissolution occurred during the simulation process, which would increase the $H^+$ concentration in the groundwater, decrease the pH value, and increase the $HCO_3^-$ concentration for the three simulation paths.

**Table 3.** Groundwater sample chemical composition of three simulated paths.

| Reaction Path | Path 1 | | Path 2 | | Path 3 | |
|---|---|---|---|---|---|---|
| Sample ID | X-17 | X-15 | Z-19 | Z-12 | D-2 | D-11 |
| pH | 7.54 | 7.35 | 7.67 | 7.39 | 7.36 | 6.99 |
| T (°C) | 23.70 | 24.20 | 21.40 | 20.70 | 19.40 | 21.10 |
| $Ca^{2+}$ (mg/L) | 143.59 | 235.89 | 92.32 | 142.72 | 80.21 | 200.51 |
| $Mg^{2+}$ (mg/L) | 24.61 | 26.92 | 27.81 | 42.32 | 25.94 | 44.75 |
| $Na^+$ (mg/L) | 58.36 | 91.69 | 45.65 | 90.44 | 65.12 | 129.82 |
| $K^+$ (mg/L) | 1.56 | 1.73 | 7.42 | 1.97 | 2.55 | 4.60 |
| $HCO_3^-$ (mg/L) | 225.77 | 298.99 | 146.45 | 317.30 | 176.96 | 268.49 |
| $SO_4^{2-}$ (mg/L) | 68.52 | 99.30 | 316.08 | 72.53 | 170.71 | 299.78 |
| $Cl^-$ (mg/L) | 169.73 | 277.13 | 57.27 | 289.95 | 83.50 | 379.57 |
| Si | 11.52 | 10.87 | 0.38 | 13.55 | 5.86 | 17.76 |

The Cenozoic Quaternary is the main exposed strata in the study area, and the Quaternary is mainly distributed with gypsum, dolomite, calcite, and feldspar minerals [29]. In this study, calcite, dolomite, gypsum, halite, quartz, albite, potash feldspar, and cation exchange were selected as a "possible mineral phase". According to the groundwater hydrochemistry results in the study area, eight elements of $Ca^{2+}$, $Mg^{2+}$, $Na^+$, $K^+$, $HCO_3^-$, $SO_4^{2-}$, $Cl^-$, and Si were taken as constraint variables. Putting the groundwater hydrochemical data at the starting and ending points to the PHREEQC interface and applying the selected "possible mineral phase" to obtain the optimal solution on each path, the reverse simulation results of migrated mass for each mineral phase on three paths are shown in Table 4. From the table, it can be seen that the cation exchange had taken place on three simulation paths, where $Ca^{2+}$ entered the groundwater, while a large amount of $Na^+$ was adsorbed on the particle surface in the aquifer, which was evidence of more active SI in Longkou city. Due to the relatively high concentration of $Na^+$ in the coastal groundwater, which exceeded the equilibrium concentration of exchangeable cations on the clay surface, $Na^+$ in the groundwater was replaced by $Ca^{2+}$ and $Mg^{2+}$ in the surrounding rock or soil. In Table 4, dolomite and quartz precipitated with negative migrated masses of $1.38 \times 10^{-3}$ and $1.08 \times 10^{-5}$ mol/L can be seen for simulation Path 1, while calcite, halite, and gypsum dissolved with positive migrated masses of $2.89 \times 10^{-3}$, $3.52 \times 10^{-3}$, and $4.66 \times 10^{-4}$ mol/L, respectively. This was maybe because the mixing of seawater and groundwater was considered in the whole hydrogeochemical process, and the dissolution of halite was closely related to seawater. A Gibbs diagram of groundwater in the study area is further presented in Figure 10. In the in right top area of Figure 10, seawater samples, including X-12, were located, which were within the evaporative concentration zone. This indicated the simulation Path 1 was affected by the seawater and evaporative concentration together. Furthermore, the dolomite precipitation reduced the $Ca^{2+}$ concentration, and the quartz precipitation reduced the Si concentration, while the dissolution of calcite, rock salt, and gypsum increased the concentrations of $Na^+$, $Ca^{2+}$, $SO_4^2$, and $Cl^-$ in groundwater. The $Ca^{2+}$ on Path 1 mainly came from the dissolution of minerals and cation exchange. Because the dissolution amount sum of calcite and gypsum and the adsorption amount sum of $Ca^{2+}$ in cation exchange ($2.89 \times 10^{-3} + 4.66 \times 10^{-4} + 1.45 \times 10^{-3} = 4.806 \times 10^{-3}$ mol/L) were larger than the precipitation amount of dolomite ($1.38 \times 10^{-3}$ mol/L), the $Ca^{2+}$ concentration increased from 143.59 to 235.89 mg/L in Table 3 and the $Mg^{2+}$ concentration on Path 1 increased too. From the simulation results in Table 4, the precipitation of dolomite, with an absolute migrated mass of $1.38 \times 10^{-3}$ mol/L, was larger than that of $Mg^{2+}$ in cation exchange, with a migrated mass of $1.32 \times 10^{-3}$ mol/L. Maybe the dissolution of illite and

other minerals or dolomitization in the SI process caused the $Mg^{2+}$ increase from 24.61 to 26.92 mg/L on Path 1 in Table 3. Additionally, the dissolution of halite was the largest, with a migrated mass of $3.52 \times 10^{-3}$ mol/L, followed by calcite, with a migrated mass of $2.89 \times 10^{-3}$ mol/L, while the amount of $Na^+$ adsorbed on the mineral surface, with a negative migrated mass of $2.63 \times 10^{-3}$ mol/L, was larger than that of dolomite, with a negative migrated mass of $1.38 \times 10^{-3}$ mol/L.

**Table 4.** Reverse simulation results of migrated mass for chemical element on Paths 1, 2 and 3.

| Mineral Facies | Chemical Element | Migrated Mass (mol/L) | | |
|---|---|---|---|---|
| | | Path 1 | Path 2 | Path 3 |
| dolomite | $CaMg(CO_3)_2$ | $-1.38 \times 10^{-3}$ | $-3.56 \times 10^{-3}$ | $7.76 \times 10^{-4}$ |
| calcite | $CaCO_3$ | $2.89 \times 10^{-3}$ | $6.95 \times 10^{-3}$ | $-1.77 \times 10^{-4}$ |
| halite | $NaCl$ | $3.52 \times 10^{-3}$ | $4.56 \times 10^{-3}$ | $7.82 \times 10^{-3}$ |
| gypsum | $CaSO4$ | $4.66 \times 10^{-4}$ | $-2.44 \times 10^{-3}$ | $1.35 \times 10^{-3}$ |
| quartz | $SiO2$ | $-1.08 \times 10^{-5}$ | $2.11 \times 10^{-4}$ | $1.98 \times 10^{-4}$ |
| $MgX_2$ | $MgX_2$ | $1.32 \times 10^{-3}$ | $-9.41 \times 10^{-4}$ | $8.64 \times 10^{-4}$ |
| CaX2 | CaX2 | $1.45 \times 10^{-3}$ | $3.47 \times 10^{-4}$ | $1.64 \times 10^{-3}$ |
| NaX | NaX | $-2.63 \times 10^{-3}$ | $-5.07 \times 10^{-3}$ | $-5.00 \times 10^{-3}$ |

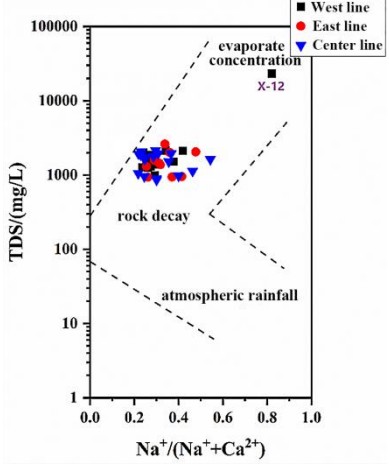

**Figure 10.** Gibbs diagram of groundwater in the study area.

Moreover, dolomite and gypsum precipitated, and calcite, halite, and quartz dissolved on simulation Path 2 from Table 4. The dissolution of halite increased the $Na^+$ and $Cl^-$ concentrations, and the precipitation of gypsum decreased the $SO_4^{2-}$ concentration in groundwater. The $Ca^{2+}$ on Path 2 also came from the dissolution of minerals and cation exchange. Because the dissolution amount sum of calcite and the adsorption amount of $Ca^{2+}$ in cation exchange ($6.95 \times 10^{-3} + 3.47 \times 10^{-4} = 7.297 \times 10^{-3}$ mol/L) was larger than the precipitation amount sum of dolomite and gypsum ($3.56 \times 10^{-3} + 2.44 \times 10^{-3} = 6.00 \times 10^{-3}$ mol/L), the $Ca^{2+}$ concentration increased from 92.32 to 142.72 mg/L on Path 2. Additionally, the $Mg^{2+}$ concentration on Path 2 increased from 27.81 to 42.32 mg/L in Table 3, but dolomite precipitated. The $Mg^{2+}$ adsorption to the mineral surface in cation exchange would reduce the $Mg^{2+}$ concentration. So, the dissolution of illite and other minerals or dolomitization in the SI process may lead to the increase of $Mg^{2+}$ concentration on Path 2. From the reverse simulation results, the dissolution of calcite was the largest, with a migrated mass of $6.95 \times 10^{-3}$ mol/L, followed by halite, with a migrated mass of $4.56 \times 10^{-3}$ mol/L. The amount of $Na^+$ adsorbed, with a migrated mass of $5.07 \times 10^{-3}$ mol/L on the mineral surface, was less than that of dolomite, with a migrated mass of $3.56 \times 10^{-3}$ mol/L, and gypsum, with a migrated mass of $2.44 \times 10^{-3}$ mol/L.

Meanwhile, on simulation Path 3, the dolomite, gypsum, halite, and quartz were dissolved, and calcite was precipitated with negative migrated mass of $1.77 \times 10^{-4}$ mol/L from Table 4. Among them, the dissolution of dolomite, gypsum and halite significantly increased the $Na^+$, $Cl^-$, $Ca^{2+}$, and $SO_4^{2-}$ concentrations, from 65.12 to 129.82 mg/L, 83.5 to 379.57 mg/L, 80.21 to 200.51 mg/L, and from 170.71 to 299.78 mg/L in groundwater, respectively, while the dissolution of quartz increased the Si concentration from 5.86 to 17.76 mg/L in Table 3. The dissolved amount sum of gypsum and dolomite and the adsorption amount of $Ca^{2+}$ in cation exchange ($1.35 \times 10^{-3} + 7.76 \times 10^{-4} + 1.64 \times 10^{-3} = 3.766 \times 10^{-3}$ mol/L) was larger than that of calcite precipitation ($1.77 \times 10^{-4}$ mol/L), so the $Ca^{2+}$ concentration increased. The $Mg^{2+}$ on Path 3 came from the $Mg^{2+}$ dissolution in cation exchange and the dissolution of illite and other minerals. From the reverse simulation results in Table 4, the halite dissolution was the largest with migrated mass of $7.82 \times 10^{-3}$ mol/L, followed by gypsum, with a migrated mass of $1.35 \times 10^{-3}$ mol/L. The amount of $Na^+$ adsorbed, with an absolute migrated mass of $5.00 \times 10^{-3}$ mol/L on the mineral surface, was larger than that of calcite, with an absolute migrated mass of $1.77 \times 10^{-4}$ mol/L.

## 6. Conclusions

Based on the observed geological and hydrogeological data with 44 groundwater samples and 3 surface seawater samples, hydrochemistry analysis, correlation analysis, principal component analysis, stable isotope tracing, three reverse geochemical simulation paths, and other methods were firstly comprehensively used, and the hydrogeochemistry effect of the SI process in the study area was revealed. Main conclusions can be obtained.

The increase of TDS was mainly related to the dissolution of halite, while $Cl^-$ and $Na^+$ contributed the most to TDS in coastal areas. The high correlation coefficients of $Na^+$, $Mg^{2+}$, $K^+$, and $Cl^-$ were 0.998, 0.996, and 0.964, respectively, and they had similar sources, mainly from SI. The SI degree was the strongest at the junction of the west and north coast zones. The TDS concentration in the west coast was high, but the $Cl^-$ concentration was relatively low, indicating other ions in this area contributed much more than $Cl^-$ to TDS. The principal component one was mainly TDS, $Mg^{2+}$, $Cl^-$, $Na^+$, $SO_4^{2-}$, $K^+$, and $Ca^{2+}$, with load values of 0.994, 0.993, 0.986, 0.985, 0.942, 0.929, and 0.926, respectively, which can represent the indicator of the SI degree. The principal component two was mainly composed of $NO_3^-$ and $HCO_3^-$, with load matrix values of 0.889 and 0.483, respectively, which represented the degree of carbonate dissolution and human pollution. $Na^+$ and $Ca^{2+}$ were the main cations, while $Cl^-$ and $SO_4^2$ were the main ions. The hydrochemical types of groundwater in Longkou city were mainly $HCO_3 \cdot Cl$-Na.Ca and $HCO_3 \cdot Cl$-Ca, and the groundwater samples generally showed the evolution from $HCO_3$-Ca to $HCO_3 \cdot Cl$-Na to Cl-Na from the inland to the coastline. The main source of groundwater was atmospheric precipitation from the stable isotope analysis of $\delta^2H$, $\delta^{18}O$ and $^{87}Sr/^{86}Sr$. Due to the influence of seawater, $\delta^2H$ and $\delta^{18}O$ at the junction of the west and north coast had high values. The SI resulted in a high $Sr^{2+}$ concentration and the $Sr^{2+}$ of groundwater was mainly from the SI and groundwater–rock interaction. In the SI process, the mixing of SFW took place firstly, and then different degrees of cation exchange and mineral dissolution and sedimentation occurred. Results of reverse hydrogeochemical simulation showed dolomite and quartz precipitated, with negative migrated masses of $1.38 \times 10^{-3}$ and $1.08 \times 10^{-5}$ mol/L on simulation Path 1, respectively, where calcite, halite, and gypsum dissolved with positive migrated masses of $2.89 \times 10^{-3}$, $3.52 \times 10^{-3}$, and $4.66 \times 10^{-4}$ mol/L, respectively. This was also affected by seawater and evaporative concentration together, while dolomite and gypsum precipitated and calcite, halite, and quartz dissolved on simulation Path 2. On simulation Path 3, the dolomite, gypsum, halite, and quartz were dissolved, and calcite was precipitated, with a negative migrated mass of $1.77 \times 10^{-4}$ mol/L.

The study results can provide scientific references for the SI process, groundwater resource management, and geological disaster control in this area. However, human conditions, long-term hydrochemical monitoring, and samples from atmosphere and surface water are issues to address in the future.

**Author Contributions:** Y.W.: Conceptualization, Methodology, Investigation, Writing-Original draft preparation, Formal analysis, Validation, Software; J.T.: Supervision, Conceptualization, Methodology, Formal analysis, Resources, Data curation, Writing-Reviewing and Editing, project administration, funding acquisition; B.X.H.: Methodology, Project administration, Resources; H.D.: Project administration, Resources. All authors have read and agreed to the published version of the manuscript.

**Funding:** This work was partly supported by the National Key Research and Development Program of China (Grant 2016YFC0402805), the National Natural Science Foundation of China (Grants 42072271) and the Fundamental Research Funds for the Central Universities (Grant No. 2652018179).

**Institutional Review Board Statement:** Not applicable.

**Informed Consent Statement:** Written informed consent has been obtained from the authors (Yuxue Wang, Juxiu Tong, Bill X. Hu, Heng Dai) to publish this paper.

**Data Availability Statement:** The dataset generated and/or analyzed during the current study are not publicly available due to further dissertation writing but are available from the corresponding authors on reasonable request.

**Conflicts of Interest:** The authors declare no conflict of interest.

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
