# Peer review of "Combining Isotope and Hydrogeochemistry Methods to Study the Seawater Intrusion: A Case Study in Longkou City, Shandong Province, China"

_water, doi:10.3390/w14050789_

Round 1

Reviewer 1 Report

Comment 1: The abstract should state briefly the purpose of the research, the principal results and major conclusions. An abstract is often presented separately from the article, so it must be able to stand alone.

Comment 2: More suitable title should be presented for the figure 3 instead of “Principal component eigenvalue and variance contribution rate and load matrix.”.

Comment 3: Please underscore the scientific value added of your paper in your abstract and introduction.

Comment 4: Methods section determines the results. Kindly focus on three basic elements of the methods section.

  1. How the study was designed?
  2. How the study was carried out?
  3. How the data were analyzed?

Comment 5: The major defect of this study is the debate or Argument is not clear stated in the introduction session. Hence, the contribution is weak in this manuscript. I would suggest the author to enhance your theoretical discussion and arrives your debate or argument.

Comment 6: It is suggested to add articles entitled “Nazarnia et al. A Systematic Review of Civil and Environmental Infrastructures for Coastal Adaptation to Sea Level Rise” and “Emmy C. Kerich. Households Drinking Water Sources and Treatment Methods Options in a Regional Irrigation Scheme” to the literature review.

Comment 7: Especially, the introduction section needs to re-organize. The major debate or Argument is not clear stated in the introduction session. Hence, the contribution debates are weak in this manuscript. I would suggest the author to enhance your literature discussion and arrives your debate or argument.

Comment 8: Please explain your results into steps and links to your proposed method.

Comment 9: Please make sure your conclusions' section underscore the scientific value added of your paper, and/or the applicability of your findings/results, as indicated previously. Please revise your conclusion part into more details. Basically, you should enhance your contributions, limitations, underscore the scientific value added of your paper, and/or the applicability of your findings/results and future study in this session.

Comment 10: I would like to request the author to emphasis on the contributions on practically and academically in implication session.

Author Response

Comments and Suggestions for Authors

Comment 1: The abstract should state briefly the purpose of the research, the principal results and major conclusions. An abstract is often presented separately from the article, so it must be able to stand alone.

Reply: The summary section has been simplified and the research purpose and scientific value have been added in the abstract.

Comment 2: More suitable title should be presented for the figure 3 instead of “Principal component eigenvalue and variance contribution rate and load matrix.”.

Reply: Figure 3 has changed the title to "element principal component and load matrix".

Comment 3: Please underscore the scientific value added of your paper in your abstract and introduction.

Reply: We have rewritten the abstract and introduction section to highlight the contribution of this study in the revised manuscript as follows:

In order to study the hydrogeochemical effect in the process of seawater intrusion (SI), and provide scientific basis for comprehensive management of water resources and water ecological restoration, ……”

“Based on the observed data, traditional hydrogeochemistry methods of hydrochemistry analysis, correlation analysis, principal component, reverse geochemical simulation was firstly comprehensively combined with stable isotope tracing in Longkou city, and it is the first time to use isotope method to study SI in the study area.”

“However, the study on stable isotope characteristics has not been investigated yet in the Longkou city. Moreover, at present, the evaluation methods of SI mainly include traditional hydrogeochemistry methods, i.e., single index method, principal component, mathematical statistics method, attribute identification method, etc (Qiao et al., 2011). However, the calculation results of single index method have some defects such as uncertainty and one sidedness. The analysis results of mathematical statistics method are relatively related to the sample data, which are easy to be fluctuated, and the selection of confidence level is difficult to grasp by attribute identification method (Liu et al., 2010). This demands new investigation method into the SI process and water supply source in the Longkou city.”

“Therefore, purposes of this study were to study the SI in the Longkou city by firstly combining the isotope method and traditional hydrogeochemistry. It should be noted that this is the first time to use isotope method to study SI in the Longkou city.”

Comment 4: Methods section determines the results. Kindly focus on three basic elements of the methods section.

  1. How the study was designed?
  2. How the study was carried out?
  3. How the data were analyzed?

Reply: 1. The field sampling period was from February 28th to March 2nd, 2019. This sampling roughly follows the principle of sequential sampling from sea to land, and the sampling scope covers three areas: the west line, the middle line and the east line. This is described in section 2.2.

  1. From the Figure 2(d), 44 groundwater samples, including 14 groundwater samples on the west line, 18 groundwater samples on the middle line and 12 groundwater samples on the east line, and 3 seawater samples were collected in the study area. Among them, groundwater is mainly collected from civil wells, which is shallow groundwater, most of which are buried below 10m, and seawater is mainly collected from surface seawater. This is described in section 2.2.
  2. It has been supplemented in section 2.2 "sampling and analysis" section.

Comment 5: The major defect of this study is the debate or Argument is not clear stated in the introduction session. Hence, the contribution is weak in this manuscript. I would suggest the author to enhance your theoretical discussion and arrives your debate or argument.

Reply: We have rewritten the introduction section to highlight the contribution of this manuscript, including:

However, the study on stable isotope characteristics has not been investigated yet in the Longkou city.

However, the calculation results of single index method have some defects such as uncertainty and one sidedness. The analysis results of mathematical statistics method are relatively related to the sample data, which are easy to be fluctuated, and the selection of confidence level is difficult to grasp by attribute identification method (Liu et al., 2010). This demands new investigation method into the SI process and water supply source in the Longkou city.

Therefore, purposes of this study were to study the SI in the Longkou city by firstly combining the isotope method and traditional hydrogeochemistry. It should be noted that this is the first time to use isotope method to study SI in the Longkou city.

Comment 6: It is suggested to add articles entitled “Nazarnia et al. A Systematic Review of Civil and Environmental Infrastructures for Coastal Adaptation to Sea Level Rise” and “Emmy C. Kerich. Households Drinking Water Sources and Treatment Methods Options in a Regional Irrigation Scheme” to the literature review.

Reply: Nazarnia et al. (2020) is cited in the revised manuscript now. The second suggested reference can not be found, so we do not cite it.

Comment 7: Especially, the introduction section needs to re-organize. The major debate or Argument is not clear stated in the introduction session. Hence, the contribution debates are weak in this manuscript. I would suggest the author to enhance your literature discussion and arrives your debate or argument.

Reply: Done. The introduction section is re-organized in the revised manuscript now.

Comment 8: Please explain your results into steps and links to your proposed method.

Reply: Done. We have added some words in the first paragraph for each section to link our proposed method, including:

Figure 1. Diagrammatic sketch of flowchart to study the SI in Longkou city.

Ion correlation analysis can explain the similarity and difference of hydrochemical parameters for groundwater and their corresponding sources (Adams et al., 2001), while TDS value represents the content of dissolved matter in groundwater, which can reveal the groundwater evolution.

Cl- and TDS were considered as the main component 1 in this study, which are the most commonly used indicators for judging the SI (Chen et al., 2019). Therefore, the spatial distribution of Cl- and TDS were shown in Figure 4 to explore the SI, where the Kriging method was used for interpolation.

Since analyzing hydrochemical types of groundwater has a significance to reveal the spatial distribution and hydrogeochemical composition in groundwater (Zhang et al., 2011), chemical types and salt source would be analyzed in the study area based on the above analysis.

In order to further explore the mixing degree of groundwater and seawater and the dissolution of rocks and minerals in the study area, the ratio of each standard seawater ion concentration to Cl- concentration, named the seawater ratio was used in this study.

Stable isotopes of oxygen and hydrogen were applied to further explore the effect of groundwater by the seawater in the Longkou city. The inverse distance method of Arcgis software was used to interpolate spatial distribution of δ2H and δ18O, which was shown in Figure 7.

Groundwater sources can be obtained by the stable isotopes of δ2H and δ18O, and stable isotope of strontium can be further used to explore the interaction between the groundwater and aquifer, which was applied in this study.”

Reverse hydrogeochemical simulation is to simulate the water rock reaction on the reaction path by knowing the hydrochemical and isotopic indexes of the starting point samples on the same flow path. Therefore, reverse hydrogeochemical simulation can simulate the hydrogeochemical process on the specific path, which was further used in this study.

Comment 9: Please make sure your conclusions' section underscore the scientific value added of your paper, and/or the applicability of your findings/results, as indicated previously. Please revise your conclusion part into more details. Basically, you should enhance your contributions, limitations, underscore the scientific value added of your paper, and/or the applicability of your findings/results and future study in this session.

Reply: In the conclusion section, the scientific value and limitations of this study are added as follows:

The study results can provide scientific references for the SI process, groundwater resource management and geological disaster control in this area. However, human conditions, long-term hydrochemical monitoring and samples from atmosphere and surface water were issues should be done in future.

Comment 10: I would like to request the author to emphasis on the contributions on practically and academically in implication session.

Reply: Done. We have revised the manuscript to emphasis on the practical and academical contributions as follows:

In order to study the hydrogeochemical effect in the process of seawater intrusion (SI), and provide scientific basis for comprehensive management of water resources and water ecological restoration,……

However, the study on stable isotope characteristics has not been investigated yet in the Longkou city.

Therefore, purposes of this study were to study the SI in the Longkou city by firstly combining the isotope method and traditional hydrogeochemistry. It should be noted that this is the first time to use isotope method to study SI in the Longkou city.

The study results can provide scientific references for the SI process, groundwater resource management and geological disaster control in this area.

Reviewer 2 Report

An paper by Yuxue Wang et al. “Combining isotope and hydrogeochemistry methods to study the seawater intrusion: a case study in Longkou city, Shandong province, China” combines isotope and hydrogeochemistry methods to study the intrusion of seawater. However, the interesting material received is not presented properly, in my opinion.

  1. Materials and Methods

Figure 1 (d) is completely illegible.

Table of isotopic-chemical composition of groundwater with Samples ID; Samples date/depth, m; T (°C); pH; TDS (mg·L−1); Na+; Ca2+, etc is missing.

There is no description of the "Reverse hydrogeochemical simulation" method.

Section 5 states that “three paths were selected for simulation along the groundwater flow direction. Path 1 was on west line with sampling points of X17-X15, path 2 was on middle line with sampling points of Z19-Z12, and path 3 was on east line with sampling points of D2-D11 as seen in Figure 1 (c) ". However, this is not shown in Figure 1 (c). Apparently Figure 1 (d) was meant, but three paths are not visible on it either. It is not clear how "the groundwater flow direction" was defined, for this a hydroisohypsum map is needed. The hydroisohypsum map is used to determine the direction of movement of groundwater and to determine the value of the hydraulic gradient.

It is unclear why “halite and gypsum dissolved (Line 501)” if “The surface layer is mostly sub-clay and sub-sandy soil, and the underlying layers are of sand and gravel (Lines 153-154)”. Perhaps it is necessary to consider the possibility of mixing processes of salty sea water with fresh rainwater?

  1. Hydrochemistry

You write that “TDS had high correlation with Cl- and SO42- with corresponding values ​​of 0.999 and 0.987, while it had significant negative correlation with HCO3- as value of -0.231, indicating that the increase of TDS was mainly related to the dissolution of halite and aluminosilicate (Lines 201-203) ". What do you mean by aluminosilicates? Dissolution of aluminosilicates usually increases the concentration of HCO3-.

To analyze the evolution of the chemical composition of fresh waters with an increase in their TDS, in my opinion, the graphs (eq/eq) Ca vs SO4, Ca + Mg vs HCO3, Na vs HCO3, Na/Ca vs TDS, Na vs Cl are more informative (https://doi.org/ 10.1016 / j.scitotenv.2017.10.197).

Figure 4 and 6 are completely illegible.

It is not clear what you mean by "brine" (Figure 5 and in the text). In my opinion, "Brine" is water with a TDS of more than 50 g/l. Apparently, your "brine" is shown in the figure with crosses. Please clarify the legend.

In general, the presented discussions of the evolution of the composition of groundwater are difficult to understand without a clear hydrodynamic and hydrochemical basis in the form of hydrogeological schemes and sections.

  1. Environmental isotopes

Figure 7, 8 and 9 are completely illegible.

You do not have data on the composition of your seawater samples (see note to 2. Materials and Methods), so the discussion is not always clear.

Round 2

Reviewer 1 Report

The Revisions are satisfactory in my opinion, and I would certainly recommend the Editors to Publish the Paper in their esteemed Journal. 

Author Response

Many Thanks.

Reviewer 2 Report

The authors improved their manuscript somewhat, but the explanations presented did not satisfy me.

First, you need to submit, in addition to the draft version, the manuscript in its final form. The draft version is illegible.

Second, the following explanations needs to be further revised:

  1. Reply: We have added the explanation for why “halite and gypsum dissolved” as follows:

“This is maybe because gypsum is a kind of gravel. In addition, the mixing of seawater and groundwater is considered in the whole hydrogeochemical process, and the dissolution of halite is closely related to seawater.”

I cannot agree with this explanation. Gypsum cannot be a type of gravel, it can only form inclusions, interlayers and lenses in the host rocks. Favorable conditions must exist for this. References are needed to the literature, which describes the gypsuming of sand and gravel deposits in the study area. The dissolution of halite can also occur only if there are inclusions, interlayers and lenses of halite in the host rocks. If they are absent, then seawater plays the main role in groundwater salinization (samples X12, X19, X20). References are needed to the literature that describes marine transgressions in previous geological periods (if they existed), evaporative concentration of salts (if it took place), and so on.

  1. Reply: Thanks. We have tried to draw figure with Ca vs SO4, Ca + Mg vs HCO3, Na vs HCO3, Na/Ca vs TDS, Na vs Cl as follows:

However, the suggested figure is to analyze the evolution of the chemical composition of fresh waters with an increase in their TDS. The purpose of this study is to explore the mixing degree of groundwater and seawater and the dissolution of rocks and minerals in the study area, so we do not add the figure in the manuscript.

It was not possible to combine samples with TDS more than 3 g/l on the same graph with fresh and brackish water samples. BHA, X12, X19, X20 samples should be removed from your graphs. Rationale: The Ca vs SO4 graph makes it possible to judge the processes of gypsum dissolution, the Na vs Cl (not logCl) graph - about the processes of halite dissolution and mixing fresh water with sea water, the Ca + Mg vs HCO3 graph - about the processes of carbonate dissolution, Na vs HCO3, Na/ Ca vs TDS (not log TDS) - about the processes of dissolution of aluminosilicates and cation exchange. All this is the purpose of your study.

  1. Reply: Thanks. It is a mistake, and we have deleted “aluminosilicates” in the revised manuscript as follows:

“…….indicating that the increase of TDS was mainly related to the dissolution of halite”

The dissolution of halite can only occur if there are interbeds of halite in the host rocks. Additional justifications are needed.

  1. Reply: The legend is at the bottom of the Figure 5. We have added sentence to define “brine” and clarify the legend as follows:

“Freshwater refers to water with TDS less than 1 g/L, brackish water refers to water with TDS greater than 1 g/L and less than 10 g/L, and brine refers to water with TDS greater than 10 g/L. Different types of groundwater samples in the study area were projected onto the piper triple line map, as shown in Figure 5. In the figure, the brackish water were in the three lines, while the fresh groundwater were in the east and middle lines…….”

Based on the use of which of the existing classifications do you classify "brines" as water with a TDS greater than 10 g/l? "Brine" is water with a TDS of more than 50 g/l.

Thirdly, usually, a numerical value is recorded up to the second decimal place, if the main value is measured in units, and up to the first decimal place if the main value is measured in tens. 1.12, 11.3, 234, 1345.
